# Influence of Parent-of-Origin on Intellectual Outcomes in the Chromosome 22q11.2 Deletion Syndrome

**DOI:** 10.3390/genes13101800

**Published:** 2022-10-05

**Authors:** Daniel E. McGinn, T. Blaine Crowley, Tracy Heung, Oanh Tran, Edward Moss, Elaine H. Zackai, Beverly S. Emanuel, Eva W. C. Chow, Bernice E. Morrow, Ann Swillen, Anne S. Bassett, Donna M. McDonald-McGinn

**Affiliations:** 1The 22q and You Center, Clinical Genetics Center, Division of Human Genetics, Children’s Hospital of Philadelphia, Philadelphia, PA 19104, USA; 2The Perelman School of Medicine, University of Pennsylvania, Philadelphia, PA 19104, USA; 3Clinical Genetics Research Program, Centre for Addiction and Mental Health, Toronto, ON M5G 2C4, Canada; 4The Dalglish Family 22q Clinic, University Health Network, Toronto, ON M5G 2C4, Canada; 5Department of Psychiatry, University of Toronto, Toronto, ON M5G 2C4, Canada; 6Albert Einstein College of Medicine, Bronx, NY 10461, USA; 7The Centre for Human Genetics, University Hospital of Leuven, Department of Human Genetics, University of Leuven (KU Leuven), 3000 Leuven, Belgium; 8Toronto General Hospital Research Institute, and Campbell Family Mental Health Research Institute, Toronto, ON M5G 2C4, Canada

**Keywords:** chromosome, 22q, deletion, DiGeorge, FSIQ, familial, parent-of-origin, intellect, *de novo*

## Abstract

Learning and intellectual disabilities are hallmark features of 22q11.2 deletion syndrome. Data are limited, however, regarding influences on full-scale IQ (FSIQ). Here, we investigated possible 22q11.2 deletion parent-of-origin effects. In 535 individuals, we compared FSIQ (≥50), 481 with *de novo* and 54 with inherited 22q11.2 deletions. In the subsets with data available, we examined parent-of-origin effects on FSIQ. We used linear regression models to account for covariates. Median FSIQ was significantly higher in *de novo* vs. inherited deletions (77; range 50–116 vs. 67; range 50–96, *p* < 0.0001). Results remained significant using a regression model accounting for age at IQ testing, sex and cohort site. No significant parent-of-origin differences in FSIQ were observed for *de novo* deletions (n = 81, 63.0% maternal; *p* = 0.6882). However, median FSIQ was significantly lower in maternally than in paternally inherited familial deletions (65, range 50–86 vs. 71.5, range 58–96, respectively, *p =* 0.0350), with the regression model indicating an ~8 point decrement in FSIQ for this variable (*p* = 0.0061). FSIQ is higher on average in *de novo* than in inherited 22q11.2 deletions, regardless of parental origin. However, parent-of-origin appears relevant in inherited deletions. The results have potential clinical implications with further research needed to delineate possible actionable mechanisms.

## 1. Introduction

Chromosome 22q11.2 deletion is the most common microdeletion syndrome identified in humans, with recent prevalence estimates of 1 in 2148 live births and 1 in 1000 low-risk pregnancies [1,2,3]. While 22q11.2 deletion most often occurs as a *de novo* copy number variation (CNV) due to low copy repeats (LCR) in the region, approximately 7–10% of individuals inherit the 22q11.2 deletion from an affected parent [3,4,5,6].

The phenotypic presentation of 22q11.2 deletion syndrome (22q11.2DS) is highly heterogeneous. Common features leading to the diagnosis include congenital heart disease (CHD), palatal anomalies, immunodeficiency, hypocalcemia, cognitive deficits and treatable neuropsychiatric disorders but the severity of individual features, including intellectual and learning disabilities, varies greatly [3,4,5,7,8,9]. Approximately two-thirds of individuals have a full scale intelligence quotient (FSIQ) in the borderline to mild range (55–85), and overall mean FSIQ is most commonly in the 70–74 range [10,11,12,13]. Demographic and clinical features such as sex, CHD, hypocalcemia, and attention deficit disorder have shown no association with cognitive outcomes [10,13,14,15,16]. However, in addition to the major effect of the chromosome 22q11.2 deletion itself, there is now evidence for age, major psychotic illness, and background (e.g., mid-parental FSIQ), playing a role in FSIQ outcomes, as they do in the general population [17,18,19,20]. There are limited data, however, on the challenging-to-study areas of inheritance and parental origin of the chromosome 22q11.2 deletion; e.g., no study has reported FSIQ data for more than 11 inherited deletions [10,11,17,21]. We therefore investigated the possible impact of both inheritance and parent-of-origin of the typical 22q11.2 deletion on FSIQ in individuals with 22q11.2DS. Based on previous findings from smaller studies, we predicted that those with a *de novo* deletion would have higher FSIQ scores than those with an inherited 22q11.2 deletion. We also examined possible parent-of-origin effects on FSIQ for a subset of individuals with *de novo* deletions and available research-based data, and for those with inherited deletions.

## 2. Materials and Methods

### 2.1. Participants and Study Design

Using a retrospective design, we examined cognitive outcome data available from four large cohorts of individuals with 22q11.2DS, each under IRB approved protocols. Inclusion criteria consisted of having a molecularly confirmed (FISH/MLPA/microarray) 22q11.2 deletion, inclusive of the important developmental gene *TBX1* and molecular confirmation of the *de novo* or inherited status of the 22q11.2 deletion. Most patients identified by MLPA/microarray had the common ~3 Mb deletion (LCR22A-LCR22D), or less commonly a proximal nested deletion (LCR22A-LCR22B or LCR22A-LCR22C). Distal nested deletions (LCR22B-LCR22D or LCR22C-LCR22D) were excluded. Inclusion was restricted to those having IQ data available from a standard Wechsler assessment, with FSIQ 50 or above (as severe intellectual disability is rare in 22q11.2DS and may be related to additional variants), and no psychotic illness at time of assessment [3]. For individuals [1] with an inherited 22q11.2 deletion, in the few instances where there was more than one affected sibling within the same family, the eldest was arbitrarily selected for study.

A total of 535 individuals were included across four sites, with data on 335 patients contributed from the 22q and You Center at the Children’s Hospital of Philadelphia (CHOP) in Philadelphia, USA; 101 from the Centre for Human Genetics at the University Hospital of Leuven in Belgium; 48 from the Albert Einstein College of Medicine in New York, NY, USA; and 51 from the Dalglish Family 22q Clinic in Toronto, ON, Canada. Most individuals (n = 481, 89.9%) had a *de novo* 22q11.2 deletion (n = 241, 50.1% female; mean age 117.0, SD 71.8, months). For a subset of these patients (n = 81 of 481), the parent-of-origin of the *de novo* deletion was determined using research microarray or sequencing data available from the Albert Einstein College of Medicine or the Toronto-based group [1,17,22]. Of the remaining 54/535 patients, 38 had a maternally inherited (n = 21, 55.3% female; mean age 127.3, SD 64.1, months) and 16 a paternally inherited chromosome 22q11.2 deletion (n = 12, 75.0% female; mean age 129.8, SD 64.0, months).

### 2.2. Analyses

FSIQ scores in individuals with *de novo* vs. inherited 22q11.2 deletions were initially analyzed. Thereafter, for those with data available (n = 81 with *de novo* and n = 54 with inherited 22q11.2 deletions) we examined FSIQ scores by parent-of-origin.

Linear regression models used parental 22q11.2 deletion inheritance status, age at IQ assessment, cohort site, and sex as independent variables and FSIQ as the dependent variable. Mann–Whitney U tests were used to compare continuous variables, and ꭓ^2^ test to compare categorical variables, as appropriate.

All statistical analyses were conducted with SAS 9.4 (SAS Institute, Cary, NC, USA). We defined statistical significance as *p* < 0.05, two-tailed.

## 3. Results

As postulated, the FSIQ scores of 481 individuals with *de novo* deletions (median 77, range 50–116) were significantly higher than the n = 54 with an inherited 22q11.2 deletion (median 67, range 50–96, *p* < 0.0001; Figure 1). Results were similar using the regression model to account for other variables including age at FSIQ testing, sex and cohort site (Table 1). Specifically, the FSIQ score of individuals with an inherited 22q11.2 deletion was on average approximately 8 points lower than those with a *de novo* deletion.

There were no significant between-group differences for *de novo* and inherited deletions in sex (*p* = 0.1249) or age at FSIQ testing (*p* = 0.0534). Furthermore, there was no significant difference in the origin of inheritance of 22q11.2 deletion between the cohort sites used in the study (*p* = 0.9150).

Within the *de novo* 22q11.2 deletion subgroup where parent-of-origin data were available on a research basis, there were no significant differences in FSIQ between the 51 with a 22q11.2 deletion of maternal origin (median 75, range 52–106) and the 30 with a deletion of paternal origin (median 72.5, range 56–104, *p =* 0.6882).

Among the 54 individuals with an inherited 22q11.2 deletion and FSIQ scores, 38 (70.4%) were of maternal and 16 of paternal origin, reflecting the relative paucity of paternally inherited 22q11.2 deletions given differential reproductive fitness effects in affected men [5,23]. Of note, individuals with inherited deletions of maternal origin were found to have a lower median FSIQ than those of paternal origin (65; range 50–86 vs. 71.5; range 58–96, *p* = 0.0350; Figure 2). Results were similar using the regression model to account for age, sex and cohort site (Table 1). The FSIQ scores of those with maternal inheritance on average were on average approximately 8 points lower than those with paternal inheritance of the chromosome 22q11.2 deletion.

## 4. Discussion

Importantly, our international multicenter collaborative results confirmed historic reports of higher FSIQ scores in small samples of individuals with *de novo* compared with inherited 22q11.2 deletions [11,21]. Moreover, this study examines the potential novel association of parent-of-origin influences on FSIQ for both *de novo* and familial deletions. We found no significant differences in FSIQ scores between individuals with *de novo* 22q11.2 deletions that arose either on the maternal or paternal chromosome 22q11.2. In contrast, the median FSIQ was significantly lower in maternally vs. paternally inherited familial deletions. Although these results require interpretation with caution, given the small sample size of paternally inherited 22q11.2 deletions (n = 16) available for study, they nonetheless have potential implications for genetic counseling.

Within the familial 22q11.2 deletion subgroup, there was notable consistency in the relative prevalence of maternally inherited 22q11.2 deletions (70–74%) compared with paternally inherited deletions (26–30%), regardless of cohort site. This is consistent with previously reported sex differences in reproductive fitness favoring women with 22q11.2DS [23]. These observations, together with the differential effects on offspring FSIQ as described herein, suggest that sex differences in parental origin of inherited 22q11.2 deletions deserve further study. This will include considerations such as expected assortative mating, the degree of similarity in parental FSIQ, and potentially different effects of intra-uterine environment between maternally inherited and paternally inherited 22q11.2 deletions. A high correlation in FSIQ between parents, and between parents and their offspring, is expected from both studies of the general population and of individuals with *de novo* 22q11.2 deletions [18]. Consistent with these expected parental findings for *de novo* 22q11.2 deletions, there are preliminary data from a previous study of children with inherited 22q11.2 deletions indicating some concordance in educational attainment of both parents, i.e., one with and one without a 22q11.2 deletion, though the affected parent was not indicated [11]. Larger samples will be needed to determine if there are IQ differences between men and women with the 22q11.2 deletion who reproduce, and between these individuals and their partners, with respect to offspring FSIQ outcomes.

While the largest sample yet to be assessed with respect to issues related to inheritance and parent of origin of 22q11.2 deletions, and possible effects on FSIQ of affected offspring, the current study has limitations. Cohorts with data on FSIQ and on molecular origins of the 22q11.2 deletion remain relatively small in sample size. Large studies of patients with 22q11.2DS with FSIQ data tend to have no parental molecular data available, and vice versa. The current study was under-powered to examine additional variables. For example, the 22q11.2 deletion extent may be important, given a slightly but significantly higher FSIQ reported in individuals with nested LCR22A-LCR22B deletions [13]. It is also possible that there may be differences in effect between verbal and performance IQ that were not examined in the current study [10,13,18,24]. There are also important differences in cohorts related to differing ascertainment and other methodological issues, e.g., pediatric vs. adult aged cohorts and general genetics vs. subspecialty clinics. Future well-planned prospective studies with larger samples will be important to better understand factors that may modify cognitive outcomes in individuals with 22q11.2 deletion syndrome. Those that include prenatal and developmental data, more complete cognitive and molecular data, and within-family assessment of affected and unaffected relatives, though challenging given the usually small sibships of these families, will be the most informative.

In summary, this study confirms that on average FSIQ scores tend to be lower for individuals with an inherited, compared with a *de novo* 22q11.2 deletion. Part of this effect may be related to the disproportionate maternally compared with paternally inherited deletions. The novel finding that maternally inherited familial 22q11.2 deletions appear to be significantly associated with lower FSIQ scores as compared with paternally inherited 22q11.2 deletions represents an important consideration for genetic counseling, in the prenatal and preconception setting, for patients affected by chromosome 22q11.2 deletion syndrome, as well as their family members, including grandparents who may serve as primary care providers for their grandchildren. Potential explicatory factors, in addition to partner IQ, may include maternal comorbidities, e.g., repaired congenital heart disease hypothetically affecting fetal perfusion, hypocalcemia, inflammatory processes, psychiatric illness, and teratogens such as anti-epileptics; socioeconomics, parental level of education, level of engagement of the unaffected parent; and mitochondrial or epigenetic effects, etc. It is possible that the relatively few affected men with 22q11.2DS who reproduce may have less severe neurodevelopmental phenotypes. Caution is required, as always, in providing individual risks and predicted outcomes, especially as our data on inherited 22q11.2 deletions are insufficiently powered to make far-reaching statements. The results further emphasize the need for supports for individuals with 22q11.2DS and their children [3,23]. Finally, these observations support further study, with larger numbers of paternally inherited 22q11.2 deletions, inclusive of molecular and cognitive data for both parents, as well as for affected and unaffected offspring.

## Figures and Tables

**Figure 1 genes-13-01800-f001:**
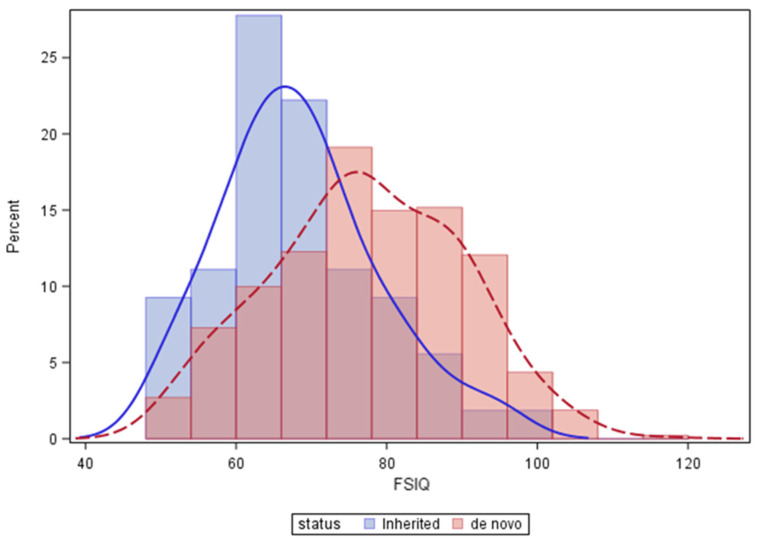
Major effect of the chromosome 22q11.2 deletion on IQ (from average 100), with further leftward shift for inherited 22q11.2 deletions compared with *de novo* 22q11.2 deletions.

**Figure 2 genes-13-01800-f002:**
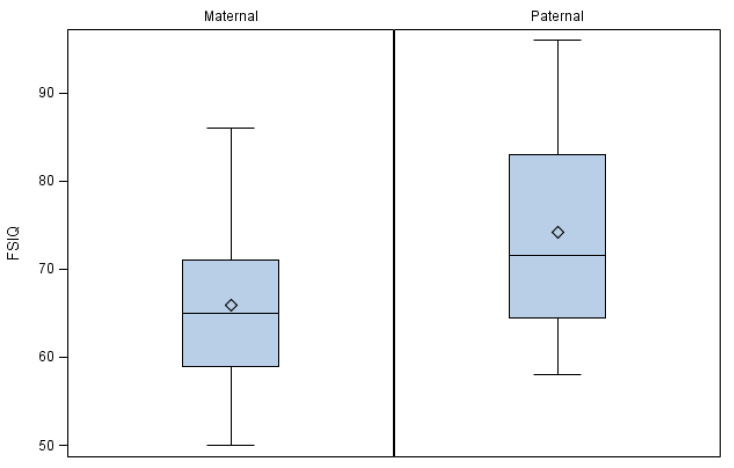
Individuals with inherited deletions of maternal origin were found to have a lower median FSIQ than those of paternal origin.

**Table 1 genes-13-01800-t001:** Linear regressions to predict offspring FSIQ by parental 22q11.2 deletion status, sex, age at IQ testing and cohort site.

Variables	Linear Regression AnalysisInherited vs. *de novo* 22q11.2 Deletion (n = 535) ^a^				Linear Regression AnalysisPaternally vs. Maternally Inherited 22q11.2 Deletion (n = 54) ^b^			
	Β	95% CI		*p*	Β	95% CI		*p*
Inherited (vs. *de novo*) 22q11.2 deletion ^a^	−8.11	−11.61	−4.62	**<0.0001**				
Paternal (vs. maternal) affected parent ^b^					8.17	2.44	13.89	**0.0061**
Age at IQ testing	−0.02	−0.03	−0.003	**0.0177**	0.05	0.003	0.10	**0.0370**
Cohort site	−1.06	−2.20	0.07	0.0655	−5.05	−8.96	−1.14	**0.0124**
Male sex	−0.95	−3.04	1.15	0.3776	−0.33	0.90	−5.84	0.9021

Β, regression coefficient; CI, confidence internal; *p* value for regression coefficient, bold values indicate statistical significance (*p* < 0.05). ^a^ The overall regression model using inherited (vs. *de novo*) status of the 22q11.2 deletion, age at IQ testing, cohort site and male sex to predict full-scale IQ was significant (adjusted R2 = 0.060, df = 4, *p* < 0.0001). ^b^ The same regression model as a, but using paternal (vs. maternal) affected parent with a 22q11.2 deletion (instead of inheritance status), was also significant (adjusted R2 = 0.256, df = 4, *p* = 0.0057). Descriptive statistics for a and b are provided in Participants and Study Design.

## Data Availability

The data presented in this study are available on request from the corresponding author. The data are not publicly available.

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
