# Peer review of "Influence of Parent-of-Origin on Intellectual Outcomes in the Chromosome 22q11.2 Deletion Syndrome"

_genes, 2022, doi:10.3390/genes13101800_

Round 1

Reviewer 1 Report

The authors discuss the intellectual outcome of individuals with 22q11.2 deletion. The study confirms that full-scale IQ (FSIQ) scores are lower in individuals with inherited 22q11.2 deletion rather than de novo deletion, particularly if maternally inherited, which is a novel finding not previously reported. The article is very well-written, and it adds to the general knowledge about 22q11.2 deletion syndrome.

There are very few points to be considered which would be beneficial for this paper.

Not clear how to determine parent-of-origin in de novo 22q11.2 deletions. It would be better to explain it in the beginning of the article, under Methods.

The result section does not mention the sex, age, school performance, life achievements (jobs/careers) of the individuals with 22q11.2 deletion.

Line 174-177: The sentence needs to be clarified further.

Line 218: The sentence starts with “And”, better to delete “and”.

Reviewer 2 Report

The results presented in this paper are very interesting. This study confirms that the individuals with inherited 22q11.2 deletions of maternal origin have a lower median FSIQ than those of paternal origin individuals. Moreover, individuals with inherited 22q11.2 deletion have lower FSIQ compared with individuals with a de novo 22q11.2 deletion.

In 81 patients with de novo deletion, the parental origin was determined using “molecular data generated on a research basis”. Could I have more information? Which technique was used?

Lines 103-105 and 151-154: these sentences should be removed.

The manuscript is clear. The methods section should be integrated. The conclusions are interesting but the small sample size of patients with inherithed deletion is not sufficient to predict patient’s outcames.
